# Resting-State Functional MRI Adaptation with Attention Graph Convolution Network for Brain Disorder Identification

**DOI:** 10.3390/brainsci12101413

**Published:** 2022-10-20

**Authors:** Ying Chu, Haonan Ren, Lishan Qiao, Mingxia Liu

**Affiliations:** 1School of Mathematics Science, Liaocheng University, Liaocheng 252000, China; 2Department of Radiology and Biomedical Research Imaging Center, University of North Carolina at Chapel Hill, Chapel Hill, NC 27599, USA

**Keywords:** domain adaptation, multi-site data, graph convolutional networks, autism, resting-state functional MRI

## Abstract

Multi-site resting-state functional magnetic resonance imaging (rs-fMRI) data can facilitate learning-based approaches to train reliable models on more data. However, significant data heterogeneity between imaging sites, caused by different scanners or protocols, can negatively impact the generalization ability of learned models. In addition, previous studies have shown that graph convolution neural networks (GCNs) are effective in mining fMRI biomarkers. However, they generally ignore the potentially different contributions of brain regions- of-interest (ROIs) to automated disease diagnosis/prognosis. In this work, we propose a multi-site rs-fMRI adaptation framework with attention GCN (A^2^GCN) for brain disorder identification. Specifically, the proposed A^2^GCN consists of three major components: (1) a node representation learning module based on GCN to extract rs-fMRI features from functional connectivity networks, (2) a node attention mechanism module to capture the contributions of ROIs, and (3) a domain adaptation module to alleviate the differences in data distribution between sites through the constraint of mean absolute error and covariance. The A^2^GCN not only reduces data heterogeneity across sites, but also improves the interpretability of the learning algorithm by exploring important ROIs. Experimental results on the public ABIDE database demonstrate that our method achieves remarkable performance in fMRI-based recognition of autism spectrum disorders.

## 1. Introduction

Resting-state functional magnetic resonance imaging (rs-fMRI) is an imaging technique that uses blood-oxygen-level-dependent (BOLD) signals to obtain functional graphs of brain activity while subjects are at rest [1]. Compared with other fMRI techniques, rs-fMRI has advantages because it is non-invasive and has high tissue resolution, and it can skillfully detect the difference between the functional activity network of the human brain under pathological conditions and that of the normal human brain [2]. At the same time, benefiting from the progress of scanning hardware and scanning technology, as well as the rapid development of computer vision technology, rs-fMRI has gradually become one of the effective means to study the human brain in recent years. Relying on rs-fMRI technology, researchers have made remarkable achievements in the auxiliary diagnosis, pathogenesis research, objective biomarker search and other aspects of mental disorders such as Autism Spectrum Disorder (ASD) and Major Depressive Disorder [3,4].

Currently, the application of machine learning/deep learning in natural image analysis is very successful. In contrast, its use in the analysis of neuroimaging data presents some unique problems, including dimensional disaster, small sample size, and limited true labels [5,6]. With the continued efforts of researchers, public multi-site neuroimage datasets, increasing the sample size and statistical power of data, are helping to promote the adoption of data-driven machine learning/deep learning techniques. However, the study of multi-site datasets will face another important challenge. That is, the distribution of data between sites is often quite different due to external factors such as different scanners or protocols [7,8]. This will severely limit the generalization ability of machine/deep learning models, as such algorithms often start with the assumption that all data remain the same distribution [9,10,11].

Studies have shown that detection of abnormal low-frequency fluctuations in the BOLD signals caused by pathological changes in the resting state will facilitate the analysis of brain connectivity and provide scientific and reliable treatment options before and after surgery [12]. Typically in studies of neuroimaging data, brain functional connectivity networks (FCNs) attempt to establish a potential causal link between two regions-of-interest (ROIs) based on linear temporal correlations [13]. Previous studies usually use statistical measures of FCNs (including betweenness centrality, degree centrality, and other features) to construct prediction models [14,15]. These practices often rely on extensive expert knowledge and are subjective, expensive, and time-consuming. FCN is usually defined as a complex non-Euclidean space graph structure [16]. In recent years, graph neural networks, especially graph convolutional networks (GCNs), have become one of the effective tools to deal with irregular graph data. GCN is a natural extension of the convolutional neural network in a graph domain [17,18]. It can be used as a feature extractor to learn node feature information and structure information end-to-end at the same time, which is the best choice for graph data learning task at present [19,20]. When GCN is naturally used to analyze rs-fMRI data, comprehensive mapping of brain FC patterns can effectively describe the functional activity of the brain [21,22]. However, existing studies usually ignore the potential contribution of different brain functional regions to the diagnosis of brain diseases, thus affecting the interpretability of the GCN model.

As shown in Figure 1, we construct a domain adaptation model with attention GCN (A^2^GCN) of multi-site rs-fMRI for ASD diagnosis. For the convenience of description, we set a known site as the source domain, and define the site to be predicted as the target domain. In this paper, we focus on the classification task of graphs. Therefore, we first construct the corresponding FCNs based on the rs-fMRI data of subjects from the source/target domains, and take the FCNs as the corresponding source/target graphs. Then, we use GCN as a feature extractor to capture the nodes/ROIs representations from the source/target graphs respectively through the graph convolution layers. In addition, the node attention mechanism is applied to explore the contribution weight of nodes/ROIs automatically. Finally, the objective function composed of multiple loss functions is jointly optimized, so as to establish a cross-domain classification model with a wider application range. We will use rs-fMRI data from the three sites (NYU, UM, UCLA) of the public ABIDE database [23] to identify ASD patients from healthy controls (HCs) to evaluate the performance of our approach.

The rest of this work is shown below: In Section 2, we briefly review the related research results of this work. In Section 3, we present our method and experimental setup. In Section 4, we introduce the data used in this work, the competing algorithms, and report the performance of different algorithms. At the same time, ablation experiments are added to investigate the contribution of key components in our proposed model. In Section 5, we discuss several extension studies related to this work and propose future related work. Finally, in Section 6, we summarize our proposed method.

## 2. Related Work

### 2.1. Graph Convolution Network for fMRI Analysis

At present, the application of deep learning framework, especially the graph convolutional networks (GCNs) model, to graph-structured data has aroused a warm response worldwide [24,25]. GCN is used to advance the feature learning of the network, which integrates the central node characteristics and graph topology information in the convolutional layer [26]. In particular, GCN has achieved impressive results in helping researchers build mathematical models for computer-assisted diagnosis of brain diseases and process and analyze neuroimaging data quickly and efficiently [27]. For example, Wang et al. [28] defined a GCN architecture based on features of fMRI for brain disorder analysis. Based on the spatiotemporal information of rs-fMRI time series, Yao et al. [29] constructed time-adaptive GCN architecture to study the periodic characteristics of the human brain. Gadgil et al. [30] focused on the short subsequence of BOLD signal, so as to construct a spatio-temporal GCN architecture and explore the non-stationary properties of FC. Traditional GCN research usually regards feature representations of each node as independently and equally. That is, they did not consider the unique contribution of each specific node/ROI to rs-fMRI analysis. In this paper, we will establish a ROI/node feature attention mechanism based on GCN to learn potential functional dependencies among brain regions, which allows us to identify those most informative brain regions for diagnosis. This will significantly improve the interpretability of GCN models for automated fMRI analysis.

### 2.2. Domain Adaptation for Brain Disorder Diagnosis

Data acquired from multiple imaging sites are correlated but distributed differently, which is a classic domain adaptation problem [31,32]. According to the latest research, domain adaptation related algorithms can be roughly summarized into two categories: (1) supervised domain adaptation. The target domain samples contain a large or small amount of label information; (2) unsupervised domain adaptation. There is no data label available for the target domain [33]. This work will focus on the problem of unsupervised domain adaptation, that is, samples from the source domain contain complete data labels, while samples from the target domain to be analyzed have no label information, which is more valuable and challenging for applications. In recent years, in order to achieve domain alignment, many cross-domain classification algorithms have been proposed, including adaptive methods based on discrepancy, adversarial learning and data reconstruction [34]. In recent years, domain adaptation technology has also achieved remarkable results in the field of medical imaging. Ingalhalikar et al. [35] coordinated multi-site neuroimaging data based on empirical Bayes formula to improve the accuracy of brain diagnostic classification. Guan et al. [32] defined a multi-site domain attention model based on deep learning for brain disease recognition. Zhang et al. [36] constructed an unsupervised domain adversarial network and established a brain disease prediction model with good classification performance. In this paper, we adopt the classical domain adaptation algorithm, that is, calculate the mean absolute error (MAE) and covariance of the source domain and the target domain at the same time, so as to guide the gradual alignment of node features learned from different domains and alleviate the domain offset problem.

## 3. Methodology

In this section, we will first describe the concepts and notation related to the unsupervised domain adaptation problem (as shown in Table 1), and then introduce our approach in detail.

### 3.1. Notation and Problem Formulation

In general, a feature space *X* of data and its marginal probability distribution P(X) will form a domain D. In this work, the source domain data from the distribution $P(X^{s})$ can be expressed as $X^{s}\in R^{M^{s}\times D^{s}}$; target domain data from distribution $P(X^{t})$ can be represented as $X^{t}\in R^{M^{t}\times D^{t}}$, where $D^{s}$ and $D^{t}$ are the feature dimension, and $M^{s}$ and $M^{t}$ are defined as the sample size in the source domain and target domain, respectively. In the unsupervised domain adaptation problem, the feature space and label space of the data from the source domain and the target domain are usually consistent, but the data distribution is different, that is, $P(X^{s})\neq P(X^{t})$. Our goal is to use the information learned from the source domain to assist in the graph classification task of a completely unmarked target domain. Our task is to build a good graph classification model for the target domain without any label based on labeled source domain.

In this article, we focus on representation learning of nodes on a graph. Therefore, we first build a graph for each subject of the source domain and target domain. A subject from the source domain is represented as a graph $G^{s}=\left(V^{s},A^{s},X^{s},Y^{s}\right)$, where $V^{s}$ represents a labeled collection of nodes in $G^{s}$, and $A^{s}\in R^{N^{s}\times N^{s}}$ represents the weighted adjacency matrix to quantify the connection strength between nodes. $N^{s}=\left|V^{s}\right|$ represents the number of nodes/ROIs of $G^{s}$. $X^{s}\in R^{N^{s}\times D^{s}}$ is the eigenmatrix of graph $G^{s}$, and the *i*-th row of $X^{s}$ is the eigenvector related to node *i*. $Y^{s}\in R^{M^{s}}$ is the label of $G^{s}$. In this paper, the label value of normal people is 0 and the category label of patients is 1. Similarly, each subject from the target domain is also defined as a graph $G^{t}=\left(V^{t},A^{t},X^{t}\right)$, which is a completely unlabeled network. $V^{t}$ is the node set. $N^{t}=\left|V^{t}\right|$ is the number of nodes/ROIs in $G^{t}$. $A^{t}\in R^{N^{t}\times N^{t}}$ is the weighted adjacency matrix. $X^{t}\in R^{N^{t}\times D^{t}}$ represents the feature matrix of $G^{t}$.

### 3.2. Proposed Method

The model A^2^GCN designed in this paper mainly includes three modules: node representation learning, node attention mechanism and domain adaptation module as shown in Figure 2. In addition, our model will be described in detail below.

#### 3.2.1. Node Representation Learning

To facilitate the classification task of downstream graphs, we use GCN to capture the node representation information on each graph.

First, we used the preprocessed BOLD signal to calculate the Pearson’s correlation coefficient (PC) between nodes on the graph, and defined it as the functional connectivity $e_{ij}\in \left[-1,1\right]$ of the *i*-th and *j*-th brain regions, as follows:(1)$$e_{ij}=\frac{{\left(v_{i}-{\bar{v}}_{i}\right)}^{⊤}\left(v_{j}-{\bar{v}}_{j}\right)}{\sqrt{{\left(v_{i}-{\bar{v}}_{i}\right)}^{⊤}\left(v_{i}-{\bar{v}}_{i}\right)}\sqrt{{\left(v_{j}-{\bar{v}}_{j}\right)}^{⊤}\left(v_{j}-{\bar{v}}_{j}\right)}}$$
where $v_{i}\in R^{ts}$, $v_{i}\in V^{s}$ or $V^{t}$, and it is the average time series signal from the *i*-th ROI. ts is the number of time points of the ROI. In addition, the ${\bar{v}}_{i}$ represents the mean vector corresponding to $v_{i}$.

Thus, for the graph, the adjacency matrix $A^{k}\in R^{N^{k}\times N^{k}}$ will be defined as:(2)$${A^{k}}_{ij}=\left\{\begin{array}{cc} 1,\; & i=j\\ \left|e_{ij}\right|,\; & otherwise \end{array}\right.$$
where *k* represents source domain *s* or target domain *t*. At the same time, for simplicity and convenience, we describe the feature matrix $X^{k}\in R^{N^{k}\times N^{k}}$, of each graph through the correlation coefficient (i.e., ${X^{k}}_{ij}=e_{ij}$).

According to the traditional GCN model, given the input feature matrix $X^{k}$ and adjacency matrix $A^{k}$, the output of the l+1-th hidden layer of the neural network *H* is:(3)$$H^{\left(l+1\right)}=\sigma \left({\tilde{D}}^{-\frac{1}{2}}A^{k}{\tilde{D}}^{-\frac{1}{2}}H^{\left(l\right)}W^{\left(l\right)}\right)$$
where ${\tilde{D}}^{-\frac{1}{2}}A^{k}{\tilde{D}}^{-\frac{1}{2}}$ is the normalization of the adjacency matrix $A^{k}$, and ${\tilde{D}}_{ii}=\Sigma_{j}{A^{k}}_{ij}$. *W* is the trainable weight matrix, that is, the parameters of the network; σ(·) is the activation function, and the ReLU function is used here. $H^{\left(l\right)}$ represents the feature matrix of the layer *l* network. l=0, then $H=X^{k}$.

#### 3.2.2. Node Attention Mechanism

For each graph, the potential impact of nodes/ROIs features learned from the GCN module on related brain diseases is different. Therefore, this paper proposes a node attention mechanism module to automatically mine the weight of nodes on the graph. See Figure 2 for details. After learning the node representation module, we naturally obtain new embedded representations of the source and target domains, that is, $H^{s}\in R^{N\times D}$ from the source domain graph and $H^{t}\in R^{N\times D}$ from the target domain graph. At this point, $N=N^{s}=N^{t}$, that is, the brains of subjects from different domains will be divided into the same number of functional areas. In addition, $D=D^{s}=D^{t}$.

Then, max pooling is performed on $H^{k}$ to generate the comprehensive representation of nodes, i.e., $H_{max}^{k}$. We send the composite node representations to the two fully connected layers respectively to automatically generate the node’s attention score, i.e., $H_{att}^{k}$, and it is defined as:(4)$$H_{att}^{k}=\sigma \left(W^{k}H_{max}^{k}+B^{k}\right)$$
where $B^{k}$ is the bias term. The dimension of hidden layer of full connection layer is *N*, and *N*. The sigmoid function as a nonlinear activation function is used to constrain each element in the range [0,1]. Among them, the ROIs that contribute more to the predicted results for the model will be assigned more weight, while the brain regions that contribute less will be assigned less weight.

Therefore, the final node representation is expressed as:(5)$$Z^{k}=H_{att}^{k}⊙H^{k}+H^{k}$$
where ⊙ represents the dot product operation, which weights the features of each extracted node.

#### 3.2.3. Domain Adaptation Module

For cross-domain classification, we propose to jointly optimize the three losses to reduce domain shift. Graph-level classification tasks typically use the readout operation to extract graph representations [37,38]. This can lead to missing important information, which can negatively affect feature alignment between domains. Therefore, we will choose to use mean absolute error (MAE) loss ($L_{M}$) and CORAL loss [39] ($L_{A}$) respectively to align features before and after the readout operation.

**MAE Loss $L_{M}$**: Considering the reality, we believe that, for the same disease and the same classification task, the node representation of the graph obtained from different domains should have a certain consistency.
(6)$$L_{M}\left(Z^{s},Z^{t}\right)=\frac{1}{N\times M\times D}\sum_{i=1}^{N}\left|{Z_{i}}^{s}-{Z_{i}}^{t}\right|$$
where $M=M^{s}=M^{t}$ is the number of samples in source or target domains.

**CORAL Loss $L_{A}$**: First, readout graph-level representations of nodes using average pooling and max pooling:(7)$$G^{k}=\frac{1}{N}\sum_{i=1}^{N}{Z_{i}}^{k}∥{\underset{i=1}{max}}^{N}{Z_{i}}^{k}$$
where ∥ denotes concatenation.

Meanwhile, CORAL loss is defined as the covariance distance of the features of source domain and target domain:(8)$$L_{A}\left(G^{s},G^{t}\right)=\frac{1}{4D^{2}}∥C_{s}-C_{t}{{∥}^{2}}_{F}$$
where ∥·∥ represents the Frobenius norm.

The covariance of source domain ($C_{s}$) or target domain ($C_{t}$) is:(9)$$C_{k}=\frac{1}{M-1}\left({{G_{i}}^{k}}^{⊤}{G_{i}}^{k}-\frac{{\left(I^{⊤}{G_{i}}^{k}\right)}^{⊤}\left(I^{⊤}{G_{i}}^{k}\right)}{M}\right)$$
where *I* is a column vector with all elements 1, and $i\in \left\{1,\cdots ,M\right\}$.

**Cross Entropy Loss $L_{C}$**. Take the cross entropy loss as the source domain classifier loss. Its objective is to minimize the classification loss of the source domain data when the data label is intact:(10)$$L_{C}\left(f_{C}\left(G^{s}\right),Y^{s}\right)=-\frac{1}{M^{s}}\sum_{i=1}^{M^{s}}{Y_{i}}^{s}log\left({\hat{Y_{i}}}^{s}\right)$$
where ${Y_{i}}^{s}$ represents the real category label of the *i*-th graph of source domain, and ${\hat{Y_{i}}}^{s}$ represents the label prediction result of the *i*-th graph of source domain. We set two fully connected layers $f_{C}$ as the label classifier for the source domain.

Finally, we obtain the overall objective function of model A^2^GCN:(11)$$L=L_{C}+\gamma_{1}L_{M}+\gamma_{2}L_{A}$$
where $\gamma_{1}$ and $\gamma_{2}$ are hyperparameters used to balance the contribution weights of $L_{C}$, $L_{M}$ and $L_{A}$.

### 3.3. Implementation

The proposed A^2^GCN model is implemented based on PyTorch platform. For fair comparison, we will use the same epoch and learning rate for all involved domain adaptation learning tasks, that is, the epoch is set to 150, the learning rate is 0.0001, and Adam is used as the optimizer to optimize the model. This A^2^GCN is composed of two layers of the graph convolution layer and two layers of the fully connected layer, and the output feature dimensions are set as 32→32→64→2. The convolution layer is nonlinearly activated using the ReLU function, and the dropout of the fully connected layer is 0.4. In order to extract more discriminative pathological features and establish a cross-domain classification model with good performance, we divided the model training into two stages. According to Equation (Equation 11), we first pre-train the node representation learning and attention mechanism module for 50 epochs. $L_{C}$ is set to 0. Both the hyperparameters $\gamma_{1}$ and $\gamma_{2}$ are set to 1. In the second stage, the above modules and category classifiers are further jointly trained for 100 epochs through Equation (Equation 11), while both the balance parameters $\gamma_{1}$ and $\gamma_{2}$ are set to 0.5.

## 4. Experiments

### 4.1. Data

To evaluate the effectiveness of our proposed approach, we use NYU, UM, and UCLA from the public Autism Brain Imaging Data Exchange (ABIDE) website (http://fcon_1000.projects.nitrc.org/indi/abide/ (accessed on 20 September 2022)) to validate our model. Meanwhile, the data from these three sites have also been used by Wang et al. [40]. Specifically, the NYU site included 164 subjects, including 71 with ASD and 93 with HC. The UM site included 113 subjects, 48 with ASD, and 65 with HC. The UCLA site included 74 subjects, 36 with ASD, and 38 with HC. We built the graph based on these three sites. The phenotypic information of the subjects involved in this study is shown in Table 2. The rs-fMRI data are from the Preprocessed Connectome Project initiative (http://preprocessed-connectomes-project.org (accessed on 20 September 2022)).

Rs-fMRI data collected at different sites will be preprocessed by a widely accepted pipeline (the Configurable Pipeline for the Analysis of Connectomes (C-PAC) [41]). The steps of preprocessing mainly include: (1) slice timing, head motion correction, (2) nuisance signal regression (ventricular, cerebrospinal fluid (CSF), white matter signal, etc.), (3) template spatial standardization of the Montreal Neurological Institute (MNI) [42], and (4) temporal filtering. Then, we use the classical AAL atlas to divide each subject’s brain into 116 functional regions and extract their average time series. Finally, each subject can generate a corresponding symmetric functional connectivity matrix based on the extracted signals, and the size of the matrix is 116×116 (according to Equation (Equation 2)). The element of the matrix represents the PC between paired ROIs.

### 4.2. Experimental Settings

In this study, we will establish a classification model through four cross-site prediction tasks: NYU→UM, NYU→UCLA, UM→NYU, UM→UCLA. The dataset before the arrow is defined as the source domain, and the dataset after the arrow is set as the target domain. The source domain samples all contained complete category labels, while the target domain subjects had no label information. Considering the limited number of samples, we will use all source/target domain samples for training and testing all target domain subjects. In order to make the result more reasonable, we repeat the training process 10 times, and take the mean value and standard deviation of each algorithm as the final result.

In this study, we will set seven metrics to evaluate the performance of the model, including: Accuracy (ACC), Precision (Pre), Recall (Rec), F1-Score (F1), Balanced accuracy (BAC), Negative predictive value (NPV), and Area under curve (AUC). The greater the value of these indexes, the better the classification performance of the model. These metrics are calculated as follows: ACC = $\frac{TP+TN}{TP+FN+FP+TN}$, Pre = $\frac{TP}{TP+FP}$, Rec = $\frac{TP}{TP+FN}$, NPV = $\frac{TN}{TN+FN}$, BAC = $\frac{TP}{2(TP+FN)}$+$\frac{TN}{2(TN+FP)}$, F1 = $\frac{2Pre\times Rec}{Pre+Rec}$. The *TN*, *TP*, *FN*, and *FP* represent True Negative, True Positive, False Negative, and False Positive, respectively.

### 4.3. Competing Methods

In this work, we compare the proposed A^2^GCN with five single-domain models: (1) Degree centrality (**DC**), (2) Feature fusion using betweenness centrality and degree centrality (**BD**), (3) Feature fusion using betweenness centrality, degree centrality, and closeness centrality (**BDC**), (4) Deep neural networks (**DNN**), and (5) Graph convolutional networks (**GCN**). At the same time, we compare A^2^GCN with three state-of-the-art domain adaptation methods: (1) Cross-domain model based on multi-layer perceptron (**DNNC**), (2) Maximum Mean Discrepancy (**MMD**), and (3) Domain Adversarial Neural Network (**DANN**). More details of these competing methods are introduced below.

(1)**DC**: This method measures the degree of nodes in the FCNs as the features of subjects. Specifically, according to Equation (Equation 2), for each subject, we can generate FCN of the size of 116 × 116, where each element in FCN is the correlation coefficient between node pairs calculated by PC. First, the degree centrality (DC) indexes of each node in the FCN are calculated. Then, the model DC takes the 116 × 1-dimensional feature vector representation obtained by computing DC for each subject as the input of the SVM classifier.(2)**BD**: This method combines the betweenness centrality (BC) and DC of nodes as the features of subjects. Based on Equation (Equation 2), the FCN of each subject is obtained, and then the BC and DC of nodes are respectively calculated. The BC and DC are concatenated into 232 × 1-dimensional vectors according to rows, used as the input of SVM.(3)**BDC**: To mitigate the lack of information or noise pollution caused by manually defined features, we further calculate the BC, DC, and closeness centrality (CC) of the node of each subject FCN. The model BDC is further sequentially splicing the DC, BC, and CC values of each subject to form a feature representation of 348 × 1-dimensional as the input of the SVM classifier.(4)**DNN**: According to the classical practice, we take the FCN of the subject in the upper triangle and pull it into a vector. In order to prevent dimensional disaster, the principal component analysis (PCA) algorithm limits the dimension of variables to 64 dimensions. Then, the features after dimensionality reduction are used as the input of model DNN. The model DNN is composed of two fully connected layers, and the output dimension is: 16→2.(5)**GCN**: GCN can combine the topological structure of the graph to deeply mine the potential information of nodes. Our A^2^GCN is inspired by GCN. Obviously, if we set $\gamma_{1}=0,\gamma_{2}=0$, A^2^GCN will crash to GCN. Similar to our proposed A^2^GCN method, first, we construct the source and target graphs, respectively, based on the FCNs of the subjects. Then, based on the source graphs, the cross entropy loss is optimized to train the classification model with good performance. Finally, the GCN model is applied directly to the target graphs to make prediction. The model GCN consists of two convolutional layers and two fully connected layers, and the output dimension is: 32→32→64→2.(6)**DNNC**: We transform our A^2^GCN model feature extractor GCN into multi-layer perceptron (MLP) to construct a simple cross-domain classification model. The model inputs are the same as the settings for the DNN model above. The output dimension of the network is set to 32→2. At the same time, add CORAL loss minimization domain offset. The covariance between the sample features of the source domain and the target domain is defined as CORAL loss. Meanwhile, CORAL loss can minimize the domain offset without additional parameters. This method is basic and efficient, and it is also one of the losses used in our A^2^GCN.(7)**MMD**: The Maximum Mean Discrepancy (MMD) method aims to reduce differences of the domain distribution by MMD. This deep transfer model uses the GCN as a feature extractor. MAE loss and CORAL loss in our model are replaced by the MMD loss [9]. Then, the two-layer MLP is used as a category classifier for MMD. The number of neurons in the output layer of convolution layer and fully connected layer is consistent with our A^2^GCN method. The reference code (https://github.com/jindongwang/transferlearning (accessed on 20 September 2022)) is publicly available.(8)**DANN**: The Domain Adversarial Neural Network (DANN) [43] is a domain adaptive method based on confrontational learning. The DANN method uses a gradient inversion layer (GRL) as $Q_{\lambda }\left(x\right)=x$ with a reversal gradient $\frac{\partial Q_{\lambda }}{\partial x}=-\lambda I$ to train a domain classifier. The adaptation parameter λ of GRL refers to [43,44]. Here, *x* represents the representation of the extracted graph. The two-layer fully connected layer is used as the domain classifier of DANN to establish the adversarial loss. The hidden layer dimension is set to 64→2; the dropout is 0.4, and ReLU is responsible for nonlinear activation. Then, the two-layer MLP is used as a category classifier for DANN. Dimensions of the output layer of the convolution layer or fully connected layer are consistent with A^2^GCN.

Note that the three conventional machine learning methods (i.e., DC, BD, and BDC) and two deep learning methods (i.e., DNN and GCN) are single-domain approaches, while the three deep learning methods (i.e., DNNC, MMD, and DANN) are state-of-the-art domain adaptation methods for cross-domain classification.

### 4.4. Results

The quantitative results of the A^2^GCN and several competing methods in ASD vs. HC classification will be reported in Table 3. We observe the following interesting findings.

(1)The four cross-domain classification models (i.e., DNNC, MMD, DANN, and A^2^GCN) achieved better results in most cases compared with several single-domain classification models (i.e., DC, BDC, DNN, and GCN). This means that the introduction of domain adaptation learning module helps to enhance the classification performance of the model, which may benefit from the transferable feature representation across sites learned by the model.(2)Graph-based (i.e., GCN, MMD, DANN, and A^2^GCN)) methods usually produce better classification results than traditional classical methods based on manually defined node features (i.e., DC, BD, and BDC) and network embeddings (i.e., DNN and DNNC). Because these traditional methods only consider the characteristics of nodes, however, those methods that use GCN as feature extractors can update and aggregate the features of nodes on the graph end-to-end with the help of the underlying topology information of FCNs, in order to learn more discriminative node representation, which may be more beneficial for ASD auxiliary diagnosis.(3)The experimental results of the proposed A^2^GCN consistently outperform all competing methods. This indicates that A^2^GCN can achieve effective domain adaptation and reduce data distribution differences, thus improving the robustness of the model.(4)Compared with three advanced cross-domain methods (i.e., DNNC, MMD, and DANN), our proposed A^2^GCN method has a competitive advantage in various domain adaptation tasks. This may be because our method adds node attention mechanism modules, which can make intelligent use of different contributions of brain regions. Meanwhile, our method adopts MAE loss and CORAL loss to align different domains step by step. These operations can partially alleviate the negative effects of noisy areas.

### 4.5. Ablation Study

The proposed A^2^GCN contains two key components, namely, node attention mechanism module and domain adaptation module. To evaluate the contribution of these two parts, we compare the proposed A^2^GCN with its three variants:(1)**A^2^GCN_A**: Similar to the A^2^GCN method, firstly, the source graph and the target graph are respectively constructed based on the subject’s FCNs. Then, the node representation on the source graph is learned based on GCN. At the same time, the node attention mechanism model mentioned in Section 3.2.2 is added to set different weight values for different nodes/brain regions of the source graph. Then, cross entropy is used to calculate the classification loss. Finally, the model trained in the source domain is applied to the prediction of the target domain graph.(2)**A^2^GCN_M**: First, based on the subject’s FCNs, the model constructs the source graph and the target graph respectively. Then, according to the node representation learning module in Section 3.2.1, the node features on the source graph and the target graph are simultaneously learned based on GCN. Then, the node attention mechanism module in Section 3.2.2 is added, and the weighted node features are used to calculate the MAE loss between domains (domain adaptation module). Finally, the cross entropy is used to calculate the classification loss.(3)**A^2^GCN_C**: First, the model uses FCNs to construct source and target graphs. Like A^2^GCN, this model learns the node features of different domains based on GCN according to the node representation learning module in Section 3.2.1. Then, after the readout operation, the CORAL loss (domain adaptation module) between domains is calculated based on the extracted graph representation vector. The cross entropy is used to calculate the classification loss of the source domain.

In Figure 3, we report the corresponding ACC and AUC values. As shown in Figure 3, we can find that the performance of three variants A^2^GCN_A (without domain adaptation module), A^2^GCN_M (with attention mechanism module and part of domain adaptation module), and A^2^GCN_C (without domain attention mechanism module) are significantly degraded in the corresponding transfer learning task. In particular, A^2^GCN_A achieved the worst performance in most cases. The underlying reason could be that attention mechanisms play a role in extracting more discriminative features. In addition, it also shows that using MAE loss and CORAL loss to align the learned features step by step during training can reduce the data information loss caused by readout-related pooling operations, thus significantly improving the robustness and transmission performance of A^2^GCN. More results on the influence of parameters and model pre-training can be found in Appendix A.

## 5. Discussion

### 5.1. Visualization of Data Distribution

To visually demonstrate the features learned through the proposed A^2^GCN, we use the t-SNE [45] tool to visualize the data distribution of different imaging sites before and after domain adaptation. In Figure 4, the blue and red dots represent the source and target domains, respectively. To visualize the regional heterogeneity before domain adaptation, we flattened the upper triangle of the FCN matrix for each sample of each site. The vector representation is obtained, which is further reduced to 64 dimensions by the PCA method as the original representation of the sample. From Figure 4a, we can observe that there is a significant domain shift between the distribution of the source domain and the target domain. We use the t-SNE algorithm to visualize feature distribution of the source and target domains after the feature extractor GCN in different cross-site classification tasks (through A^2^GCN), with results reported in Figure 4b. In Figure 4b, red and blue dots are closely clustered together. This means that the distributions of the node representations of the two domains learned by our method are close, and the domain heterogeneity has been substantially reduced. At the same time, we calculated the Frobenius norm of the covariance (CF) between samples in the source domain and the target domain, which is used to measure the difference of data distribution between different sites. It is observed that the CF between different sites is significantly reduced after domain adaptation. These results show that A^2^GCN can effectively extract transferable features and reduce domain shift.

### 5.2. Most Informative Brain Regions

One of the main focuses of this work is to use interpretable deep learning algorithms to discover the underlying differences between ASD and HC subjects. An interesting question is to identify the most informative brain regions for ASD detection. In the task of “NYU→UM”, we randomly select 10 subjects from the UM site. We then extract the features of these subjects after the attention mechanism module, select 19 brain regions with strong correlation, and visualize them using BrainNet [46] tool, with results shown in Figure 5. In Figure 5, the color of brain regions is randomly assigned, and the stick-like connections between brain regions indicate strong FC between them. For ASD vs. HC classification, we find that the most informative brain regions include the hippocampus, parahippocampal gyrus, putamen lentiform, and the vicinity of thalamus, which is also consistent with previous studies [47,48]. It validates the potential application value of our model in the discovery of rs-fMRI biomarkers for ASD identification, thus helping to improve the interpretability of learning algorithms in automated brain disease detection.

### 5.3. Limitations and Future Work

Although our proposed A^2^GCN method has achieved good results in the prediction of ASD, there is still challenging work to be considered in the future. *First*, in our current work, only knowledge transfer between a single source domain and a target domain is considered. It is also interesting to explore the shared features of multiple source domains to reduce the heterogeneity of data and thus improve the learning performance of the target domain. *Second*, the size of the training sample is relatively small. We hope to add unlabeled samples from other public datasets to assist in pre-training the proposed network in a semi-supervised learning manner, aiming to further improve model generalization capability [49].

## 6. Conclusions

In this paper, we construct a multi-site unsupervised rs-fMRI domain adaptation framework (A^2^GCN) with an attention mechanism for ASD diagnosis. The framework automatically extracts rs-fMRI features from brain FCNs with the help of the GCN model. The attention mechanism is used to explore the contribution of different brain regions to the automatic detection of brain diseases and explore the interpretable features of brain regions. In addition, our method explores mean absolute error and covariance-based constraints to alleviate data distribution differences among imaging sites. We evaluate our proposed method using rs-fMRI data from a real multi-site dataset (ABIDE). Experimental results show that the A^2^GCN has significant advantages over several advanced methods.

## Figures and Tables

**Figure 1 brainsci-12-01413-f001:**
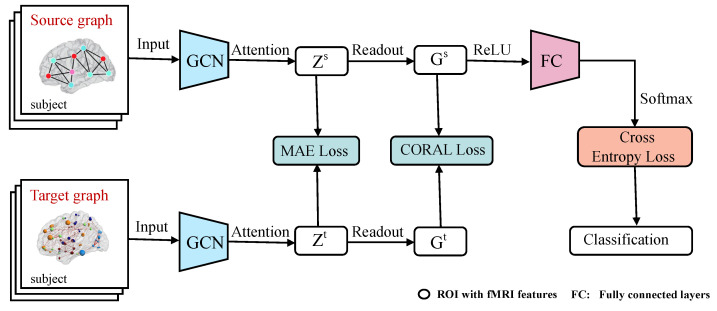
Architecture of the proposed multi-site resting-state fMRI adaptation framework (A^2^GCN) with an attention-guided GCN for brain disorder identification. The A^2^GCN consists of three components: (1) With the help of GCN model, rs-fMRI features are automatically extracted from the brain graph from the source or target domains; (2) Explore the potential contribution of different brain regions to automatic detection of brain diseases by using attention mechanism; (3) Under the constraints of mean absolute error and covariance, the objective function (composed of MAE loss, CORAL loss and cross entropy loss) is established for knowledge transfer between different domains.

**Figure 2 brainsci-12-01413-f002:**
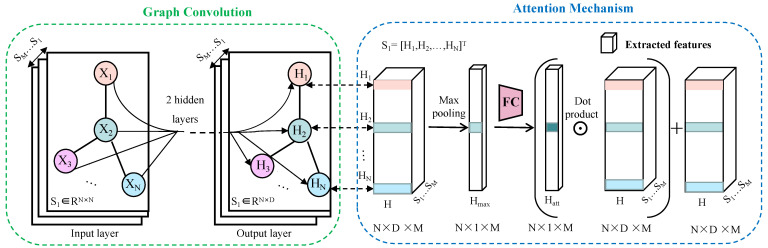
Structure of node attention mechanism module. $X_{i}\in R^{N}$ and $H_{i}\in R^{D}$ are the input and output of the convolutional layer, respectively. i=1,⋯,N. After the two-layer graph convolution, the spatial dimension of the output layer is limited to N×D×M. With the help of the max pooling operation, the global feature descriptor ($H_{max}$) of N×1×M is generated from the tensor, and then it is mapped into an attention score ($H_{att}$) through the fully connected layer, and the dimension is unchanged. Dot product this attention score with the original N×D×M tensor ($H=[S_{1},S_{2},\cdots ,S_{M}]$). The result of the dot product is added to the original N×D×M tensor (*H*), and finally each node gets the feature with the attention mechanism reweighting. FC: Fully connected layers.

**Figure 3 brainsci-12-01413-f003:**
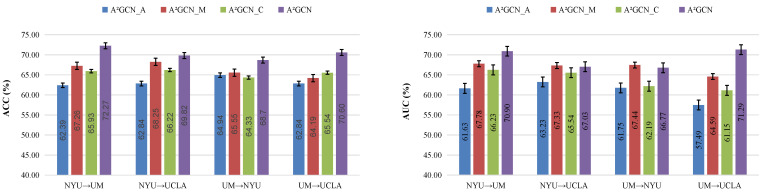
Ablation studies are performed to verify the effect of different components in the proposed model. A^2^GCN_A (without domain adaptation module), A^2^GCN_M (with attention mechanism module and part of domain adaptation module), and A^2^GCN_C (without domain attention mechanism module) are three variations of our model. ACC: Accuracy; AUC: Area under curve.

**Figure 4 brainsci-12-01413-f004:**
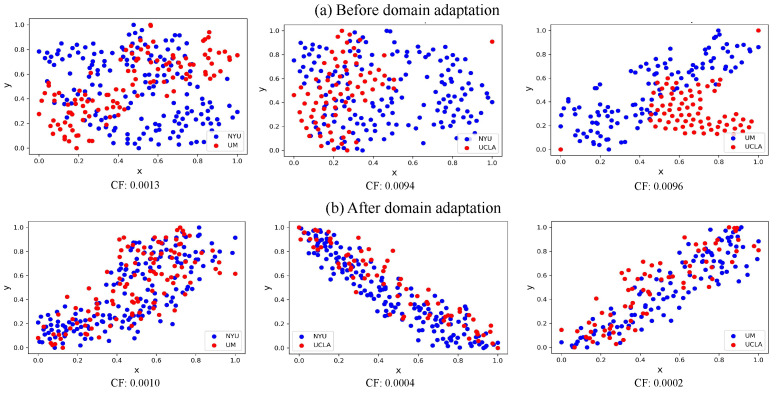
Visualization of (**a**) the original data distribution before domain adaptation and (**b**) the data distribution after adjustment through our proposed domain adaptation model for ABIDE data set. The blue dots are from the source domain and the red dots are from the target domain. CF: Frobenius norm of the covariance between the source and target domains.

**Figure 5 brainsci-12-01413-f005:**
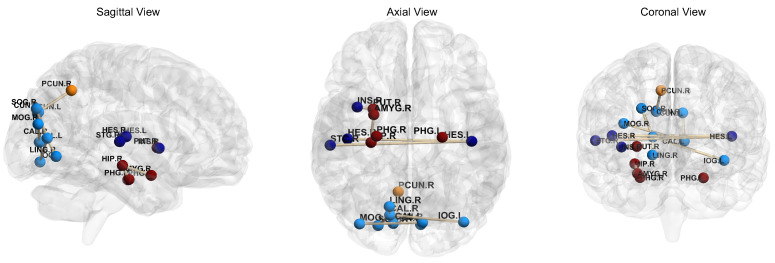
Visualization of the 19 brain regions generated by 10 randomly selected subjects from the UM site (according to the results of A^2^GCN in the domain adaptation task of “NYU→UM”). Colors of brain regions are randomly assigned, just for better visualization. The stick-like connections between brain regions indicate strong functional connectivity between them.

**Table 1 brainsci-12-01413-t001:** Notations and descriptions used in this paper.

Notation	Description
$G^{s}=\left(V^{s},A^{s},X^{s},Y^{s}\right)$	Source graph
$G^{t}=\left(V^{t},A^{t},X^{t}\right)$	Target graph
$V^{s},V^{t}$	Set of nodes
$Y^{s}\in R^{M^{s}}$	Source data label
$A,A^{s},A^{t}$	Adjacency matrix
$X^{s}\in R^{M^{s}\times D^{s}}$	Source feature matrix
$X^{t}\in R^{M^{t}\times D^{t}}$	Target feature matrix
$H^{s},H^{t}$	Learned features
$Z^{s},Z^{t}$	Learned features
$M,M^{s},M^{t}$	Number of samples
$N,N^{s},N^{t}$	Number of nodes on the graph
$D,D^{s},D^{t}$	Feature dimension
$f_{C}$	Source domain classifier
$L_{C},L_{M},L_{A}$	Loss function
$\gamma_{1},\gamma_{2}$	The balance parameters

**Table 2 brainsci-12-01413-t002:** Demographic information of three sites (NYU, UM, UCLA) of the public ABIDE dataset. Values are counted as mean ± standard deviation. M/F: Male/Female; ASD: Autism Spectrum Disorder; HC: Healthy Controls.

Name of the site	Category	Gender (M/F)	Age
NYU	ASD (N = 71)	66/5	17.59 ± 7.84
	HC (N = 93)	79/14	16.49 ± 7.68
UM	ASD (N = 48)	43/5	17.05 ± 8.36
	HC (N = 65)	56/9	17.35 ± 7.12
UCLA	ASD (N = 36)	28/8	16.27 ± 6.48
	HC (N = 38)	31/7	14.65 ± 4.97

**Table 3 brainsci-12-01413-t003:** Results of different models in ASD vs. NC classification task based on rs-fMRI data in NYU, UM, and UCLA sites. The data set preceding the arrow represents the source domain, and the arrow is followed by the target domain to predict. Values are reported as mean ± standard deviation. DC: Degree centrality; BD: Feature fusion using betweenness centrality and degree centrality; BDC: Feature fusion using betweenness centrality, degree centrality, and closeness centrality; DNN: Deep neural networks; GCN: Graph convolutional networks; DNNC: Cross-domain model based on multi-layer perceptron; MMD: Maximum Mean Discrepancy; DANN: Domain Adversarial Neural Network; ACC: Accuracy; Pre: Precision; Rec: Recall; F1: F1-Score; BAC: Balanced accuracy: NPV: Negative predictive value; AUC: Area under curve. The bold values mean to highlight the experiment results.

Source→Target	Method	ACC (%)	Pre (%)	Rec (%)	F1 (%)	BAC (%)	NPV (%)	AUC (%)
	DC	53.54 ± 1.88	46.33 ± 0.25	54.60 ± 1.89	50.55 ± 9.32	54.17 ± 1.45	62.86 ± 4.04	54.60 ± 1.89
	BD	56.64 ± 1.25	49.29 ± 1.01	58.17 ± 0.23	57.00 ± 0.90	58.09 ± 0.52	67.06 ± 0.54	58.17 ± 0.23
	BDC	54.43 ± 1.87	47.48 ± 1.55	56.51 ± 2.12	56.17 ± 2.07	56.3 ± 2.02	65.54 ± 2.69	56.51 ± 2.12
	DNN	58.85 ± 0.62	58.67 ± 1.60	58.78 ± 1.70	58.39 ± 1.17	58.78 ± 1.70	65.99 ± 2.73	51.72 ± 4.19
NYU→UM	GCN	61.07 ± 1.25	60.65 ± 0.95	60.84 ± 0.89	60.61 ± 1.05	60.84 ± 0.89	67.49 ± 0.35	59.28 ± 0.02
	DNNC	61.07 ± 1.25	61.36 ± 3.49	61.11 ± 3.59	60.27 ± 2.35	61.11 ± 3.59	69.10 ± 6.80	59.89 ± 9.31
	MMD	66.82 ± 0.63	66.20 ± 0.69	66.04 ± 0.46	66.09 ± 0.45	66.12 ± 0.35	71.32 ± 0.16	65.77 ± 1.32
	DANN	66.82 ± 0.63	66.72 ± 0.20	67.07 ± 0.16	66.56 ± 0.45	65.19 ± 2.51	70.61 ± 5.57	64.35 ± 0.84
	**A^2^GCN** (Ours)	**72.27 ± 0.51**	**71.94 ± 0.50**	**72.35 ± 0.52**	**71.97 ± 0.49**	**72.35 ± 0.52**	**78.23 ± 0.97**	**70.90 ± 1.53**
	DC	58.79 ± 2.86	57.51 ± 2.76	58.77 ± 2.88	57.92 ± 3.33	58.78 ± 2.89	60.03 ± 3.02	58.77 ± 2.88
	BD	56.08 ± 2.87	55.04 ± 2.97	56.02 ± 2.89	53.89 ± 3.47	56.00 ± 2.89	56.99 ± 2.81	56.02 ± 2.89
	BDC	58.79 ± 0.95	57.74 ± 0.84	58.75 ± 0.97	57.34 ± 1.41	58.74 ± 0.98	59.75 ± 1.10	60.11 ± 0.96
	DNN	60.14 ± 0.95	60.11 ± 0.96	60.05 ± 0.88	60.03 ± 0.85	60.05 ± 0.88	60.76 ± 0.32	59.83 ± 1.91
NYU→UCLA	GCN	61.49 ± 0.95	61.50 ± 1.00	61.44 ± 1.09	61.40 ± 1.08	61.44 ± 1.09	62.44 ± 2.06	58.19 ± 1.76
	DNNC	60.81 ± 3.82	60.88 ± 3.92	60.60 ± 3.83	60.46 ± 3.85	60.60 ± 3.83	60.47 ± 3.29	53.77 ± 3.98
	MMD	66.89 ± 0.96	66.94 ± 0.85	66.92 ± 0.88	66.88 ± 0.94	66.92 ± 0.88	68.50 ± 0.11	64.51 ± 1.91
	DANN	66.90 ± 0.95	67.14 ± 1.34	66.96 ± 1.14	66.82 ± 0.93	66.96 ± 1.14	69.28 ± 3.68	65.87 ± 0.52
	**A^2^GCN** (Ours)	**69.82 ± 1.56**	**70.09 ± 1.56**	**69.83 ± 1.56**	**69.71 ± 1.56**	**69.83 ± 1.56**	**71.38 ± 1.56**	**67.03 ± 1.56**
	DC	53.66 ± 0.86	46.31 ± 1.41	52.66 ± 1.41	45.62 ± 3.76	52.65 ± 1.46	59.00 ± 1.41	52.66 ± 1.41
	BD	57.02 ± 0.43	50.33 ± 0.46	56.45 ± 0.68	51.53 ± 1.66	56.52 ± 0.73	62.59 ± 0.89	56.46 ± 0.67
	BDC	53.66 ± 0.86	47.23 ± 0.91	54.46 ± 1.27	53.06 ± 2.11	54.48 ± 1.24	61.70 ± 1.65	54.46 ± 1.27
	DNN	59.15 ± 1.73	58.57 ± 1.74	58.65 ± 1.75	58.59 ± 1.75	58.65 ± 1.75	64.45 ± 1.58	55.49 ± 2.08
UM→NYU	GCN	63.11 ± 0.43	62.96 ± 0.03	63.15 ± 0.09	62.83 ± 0.20	63.15 ± 0.09	69.27 ± 1.03	64.35 ± 0.24
	DNNC	60.68 ± 1.29	59.99 ± 1.65	60.00 ± 1.85	59.95 ± 1.73	60.00 ± 1.85	65.49 ± 2.21	62.68 ± 3.57
	MMD	66.16 ± 1.29	65.44 ± 1.34	65.08 ± 1.26	65.18 ± 1.27	65.08 ± 1.26	69.04 ± 0.94	66.18 ± 2.17
	DANN	66.16 ± 0.43	65.59 ± 0.67	65.50 ± 1.09	65.47 ± 0.90	65.50 ± 1.09	70.14 ± 2.05	65.34 ± 0.69
	**A^2^GCN** (Ours)	**68.70 ± 0.70**	**68.73 ± 0.63**	**69.07 ± 0.65**	**68.56 ± 0.68**	**69.07 ± 0.65**	**75.52 ± 0.71**	**66.77 ± 0.43**
	DC	54.73 ± 0.95	53.81 ± 0.68	54.65 ± 1.00	51.00 ± 3.56	54.57 ± 1.09	55.48 ± 1.32	54.65 ± 1.00
	BD	54.73 ± 0.96	53.28 ± 1.10	54.80 ± 0.86	55.03 ± 0.33	54.79 ± 0.88	56.32 ± 0.62	54.80 ± 0.86
	BDC	56.08 ± 4.78	54.39 ± 4.40	56.21 ± 4.84	56.93 ± 5.09	56.18 ± 4.81	58.03 ± 5.28	56.21 ± 4.84
	DNN	56.76 ± 3.83	56.79 ± 4.02	56.69 ± 4.09	56.47 ± 4.19	56.69 ± 4.09	58.16 ± 5.10	52.31 ± 1.39
UM→UCLA	GCN	61.49 ± 0.95	61.47 ± 0.93	61.44 ± 0.88	61.43 ± 0.88	61.44 ± 0.88	62.33 ± 0.24	58.52 ± 1.50
	DNNC	60.14 ± 0.95	60.13 ± 0.96	60.05 ± 1.09	60.00 ± 1.14	60.05 ± 1.09	60.84 ± 1.87	46.50 ± 4.86
	MMD	65.54 ± 0.96	65.54 ± 0.97	65.50 ± 1.03	65.49 ± 1.03	65.50 ± 1.03	66.29 ± 1.82	65.24 ± 1.60
	DANN	65.54 ± 0.96	65.57 ± 0.93	65.57 ± 0.93	65.54 ± 0.95	65.57 ± 0.93	67.12 ± 0.64	61.26 ± 4.45
	**A^2^GCN** (Ours)	**70.61 ± 2.56**	**71.71 ± 3.42**	**70.65 ± 2.20**	**70.22 ± 2.23**	**70.52 ± 2.29**	**70.92 ± 3.09**	**71.29 ± 1.29**

## Data Availability

All data used in this study are available from the corresponding author on reasonable request.

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
