# Peer review of "Resting-State Functional MRI Adaptation with Attention Graph Convolution Network for Brain Disorder Identification"

_brainsci, 2022, doi:10.3390/brainsci12101413_

Round 1

Reviewer 1 Report

In this paper authors address brain disorder identification from rs-fMRI data acquired at multiple sites using an attention graph convolution network. If the paper present good results, many points need to be improved.

Positive

Authors use data from multiple sites and show a significant increase in performances (table 3), as well as the effectiveness of domain adaptation (figure 4).

To improve

Major

 - Readers are lost between methods comparison and ablation studies. This need to be clarified and separated.

 - It does not sound fair to compare non deep learning and deep learning based methods.

 - Following the previous point, the number of parameters/weights to train, the FLOPS, and the training time should be provided.

 - GCN explanations must be improved.

 - Section 5.1 needs to be rewritten to provide more details/explanations. Each element of the network should be discussed in details and its impact analyzed.

 - Overall english needs a deep effort. Many typos, many sentences need to be rephrased and the grammar must be checked. The flow can be also improved as information sometimes seem to have been tossed in the paper.

Minor

 - Equations 9 and 10 are redundant.

 - DNN seems to be simplistic, and as a consequence does not provide a fair comparison.

 - It has been long known that deep learning based features yield better results, but they have also limitations that are not mentioned.

 - Results in figure 4 should be completed using a metric for quantification purposes.

 - Figures 6 and 7 can be moved to complementary material or their results can be simply discussed.

 - section 5.6 Are authors referring to the potential use of semi-supervised learning? Authors intents need to be clarified.

Author Response

Response to Reviewer 1

In this paper authors address brain disorder identification from rs-fMRI data acquired at multiple sites using an attention graph convolution network. If the paper present good results, many points need to be improved.

 Positive: Authors use data from multiple sites and show a significant increase in performances (table 3), as well as the effectiveness of domain adaptation (figure 4).

 We further standardized the English grammar in the paper as suggested. Also, check all references to see if they relate to the content of the manuscript. We have marked the revision track of the paper in LaTeX in blue font so that editors and reviewers can easily view it.

Major Point 1:  - Readers are lost between methods comparison and ablation studies. This need to be clarified and separated.

Response: Thank you very much for your advice. Comparing our method with the current classical and advanced algorithms is to verify the current effectiveness and practical value of our model. The ablation study is to investigate whether the method we proposed, the important components of it, work. As suggested, we added experiments to the ablation study in Section 4.5 of the paper to investigate the contribution of the attention module and domain adaptation module, two key components in our proposed model. In this section, we reorganize the analysis of the results of the ablation experiments. In addition, in Section 4.4 of the paper, we further reorganize the analysis of the experimental results of our model and its competitive algorithm for more clarity. See the resubmitted manuscript for details.

Major Point 2 - It does not sound fair to compare non deep learning and deep learning based methods.

Response: Thank you very much for pointing this out. The traditional machine learning methods involved in this paper (i.e., DC, BD and BDC) use manually defined features as inputs and the SVM as classifier, while deep learning methods extract fMRI features automatically from data for building diagnostic models.

  In the experiments, besides the three conventional machine learning methods (i.e., DC, BD and BDC), we also compared our method with five deep learning methods, including (1) Deep neural networks (DNN), (2) Graph convolutional network (GCN), (3) Cross-domain model based on multi-layer perceptron (DNNC), (4) Margin Disparity Discrepancy (MDD), and (5) Domain Adversarial Neural Network (DANN). In particular, two deep learning methods (i.e., DNN and GCN) are single-domain approaches, while the other three deep learning methods (i.e., DNNC, MDD, and DANN) are state-of-the-art domain adaptation methods for cross-domain classification.  

  For clarity, we have now rephrased the related presentation for introducing details of these competing methods in Section 4.3 of the revised manuscript, which can also be found below.

“Note that the three conventional machine learning methods (i.e., DC, BD and BDC) and two deep learning methods (i.e., DNN and GCN) are single-domain approaches, while the three deep learning methods (i.e., DNNC, MDD, and DANN) are state-of-the-art domain adaptation methods for cross-domain classification.”

Major Point 3 - Following the previous point, the number of parameters/weights to train, the FLOPS, and the training time should be provided.

Response: Thanks for the great suggestion. The proposed A2GCN is implemented on Pytorch, with a GPU (NVIDIA GeForce RTX with 8 GB memory). The floating point of per second (FLOPS) is a measure of hardware performance, and in neural networks, we usually use floating point of operations (FLOPs) to measure the complexity of the algorithm/model. Therefore, taking the transfer learning task of “NYU→UM” as an example, we calculate the total training time of the model, FLOPs and the number of parameters in the model training phase, as shown in the following Table S1.

  From Table S1, we can see that with the increase of model complexity, the number of parameters of deep learning models increases and the training time is also prolonged. The computational complexity of the three machine learning algorithms (i.e., DC, BD and BDC) using the SVM classifier is O(dn2), where d represents the dimension of input features, and n represents the number of training samples. The computational complexity in terms of FLOPs of these machine learning algorithms using the SVM classifier is O (n2).

Table S1 Parameter and computation time comparison between the proposed method and eight competing methods in the task of  “NYU→UM” . MB: Megabytes; S: Second. FLOPs : floating point of operations.

Method

Parameters/weights to train  

Training time (S)

FLOPs

DC

O(116×1642)

20

O(1642 )

BD

O(232×1642 )

20

O(1642 )

BDC

O(348×1642)

20

O(1642 )

DNN

0.346368 MB

40

0.002148 MB

GCN

0.015274 MB

320

7.326208 MB

DNNC

0.002148 MB

50

0.346368 MB

MDD

0.030548 MB

340

9.782272 MB

DANN

0.04559 MB

350

11.47456 MB

A2GCN (Ours)

0.069794 MB

360

16.153344 MB

Major Point 4: - GCN explanations must be improved.

Response: Thank you very much for your advice. As suggested, we have added a further explanation of GCN in the introduction section of the paper. Besides, we have included more details for the competing GCN method in Section 4.3, by introducing more details such as reference, network architecture, layers, and parameters.

For the convenience of review, we copy the related text in the following.

As suggested, in Section 4.3 of the resubmitted manuscript, we further detail the network architecture of our GCN:

4.3 Competing Methods

(5) GCN: GCN can combine the topological structure of the graph to deeply mine the potential information of nodes. Our A2GCN is inspired by GCN. Obviously, if we set γ1 = 0, γ2 = 0, A2GCN will crash to GCN. Similar to our proposed A2GCN method, first, we construct the source and target graphs respectively based on the FCNs of the subjects. Then, based on the source graph, the cross entropy loss is optimized to train the classification model with good performance. Finally, the GCN model is applied directly to the target graph to make prediction. The model GCN consists of two convolutional layers and two fully connected layers, and the output dimension is: 32 → 32 → 64 → 2.”

1 Introduction

FCN is usually defined as a complex non-Euclidean space graph structure [16 ]. In recent years, graph neural networks, especially graph convolutional neural networks (GCNs), have become one of the effective tools to deal with irregular graph data. GCN is a natural extension of convolutional neural network in graph domain [ 17 ,18 ]. It can be used as a feature extractor to learn node feature information and structure information end-to-end at the same time, which is the best choice for graph data learning task at present [ 19 ,20 ]. When GCN is naturally used to analyze rs-fMRI data, comprehensive mapping of brain FC patterns can effectively describe the functional activity of the brain [21 ,22 ]. However, existing studies usually ignore the potential contribution of different brain functional regions to the diagnosis of brain diseases, thus affecting the interpretability of the GCN model.”

Major Point 5:  Section 5.1 needs to be rewritten to provide more details/explanations. Each element of the network should be discussed in details and its impact analyzed.

Response: Thank you very much for your advice. In Section 4.5 of the revised manuscript, ablation experiments are reorganized according to the proposed recommendations to investigate the contributions of two key components in our proposed model, namely, the attentional mechanism module and the domain adaptation module.

  At the same time, because the ablation experiment A2GCN_A is added and the analysis of three ablation experiments is reorganized, we redrew Figure. 3 in the newly submitted manuscript.

Major Point 6: Overall english needs a deep effort. Many typos, many sentences need to be rephrased and the grammar must be checked. The flow can be also improved as information sometimes seem to have been tossed in the paper。

Response: Thank you very much for your advice. We have corrected the corresponding grammar problems in the resubmitted manuscript as suggested. For clarity, we rethought the writing process. We have placed Section 5.1 ablation study of the original text in section 4.5 of the newly submitted manuscript.

Minor Point 1:  - Equations 9 and 10 are redundant.

Response: Thank you very much for your reasonable suggestions. The original equations 9 and 10 are further interpretations of equations 8. Equations 9 and 10 write the covariance of source domain and target domain separately, which is relatively redundant. As suggested, equations 9 and 10 are now modified into a unified formula expression. It is presented as the new equation 9 in the resubmitted manuscript.

Minor Point 2: DNN seems to be simplistic, and as a consequence does not provide a fair comparison.

Response: This is helpful feedback.

  In this work, we aim to compare our method with competing methods from two aspects: (1) machine learning vs. deep learning methods, and (2) single-domain vs. cross-domain methods. Among the eight competing methods, DNN is a single-domain deep learning method. Also, the DNNC method can be treated as a cross-domain version of DNN, by adding a domain adaptation module to DNN. It can be seen from results reported in Table 3 that, in most cases, DNN outperforms the three traditional machine methods, suggesting its efficacy. On the other hand, we can see that the performance of DNNC is much better than DNN, which proves the importance of domain adaptation for improving model performance. Therefore, in this paper, we use DNN as one of the comparison methods.

Minor Point 3:  It has been long known that deep learning based features yield better results, but they have also limitations that are not mentioned.

Response: Thank you very much for your advice.

In general, traditional machine learning algorithms, such as BD, use manually predefined features to build models. These characteristics can reflect the meaning behind the network. Compared with traditional machine learning algorithms, deep learning algorithms are difficult to explain the internal logic of model operation. Therefore, in order to better show the internal characteristics of deep learning, we have done the following two works in this paper:

1) In Section 5.1, the t-SNE tool is used to show the features learned internally:

To visually demonstrate the features learned through the proposed A2GCN, we use the t-SNE [ 45] tool to visualize the data distribution of different imaging sites before and after domain adaptation.

......

 In Figure. 4 (b), red and blue dots are closely clustered together. This means that the distributions of the node representations of the two domains learned by our method are close, and the domain heterogeneity has been substantially reduced. At the same time, we calculated the Frobenius norm of the covariance (CF) between samples in the source domain and the target domain, which is used to measure the difference of data distribution between different sites. It is observed that the CF between different sites is significantly reduced after domain adaptation. These results show that A2GCN can effectively extract transferable features and reduce domain shift.

2) Important brain regions and functional connections are shown in Section 5.2:

 One of the main focuses of this work is to use interpretable deep learning algorithms to discover the underlying differences between ASD and HC subjects. An interesting question is to identify the most informative brain regions for ASD detection. In the task of "NYU→UM", we randomly select 10 subjects from the UM site. We then extract the features of these subjects after the attention mechanism module, select 19 brain regions with strong correlation, and visualize them using BrainNet [ 46 ] tool, with results shown in
Figure. 5. In Figure. 5, the color of brain regions is randomly assigned, and the stick-like connections between brain regions indicate strong FC between them. For ASD vs. HC classification, we find that the most informative brain regions include the hippocampus, parahippocampal gyrus, putamen lentiform and the vicinity of thalamus, which is also consistent with previous studies [ 47, 48 ]. It validates the potential application value of our model in the discovery of rs-fMRI biomarkers for ASD identification, thus helping improve the interpretability of learning algorithms in automated brain disease detection.

Minor Point 4: Results in figure 4 should be completed using a metric for quantification purposes.

Response: Thank you very much for your advice. In the field of domain adaptation, Frobenius norm of the covariance is a classical and commonly used metric to calculate two data distributions. As suggested, we add the calculation of Frobenius norm of the covariance between samples of source domain and target domain, with results reported in the following Table S2.

  It is observed from Table S2 that the Frobenius norm value of the covariance between samples of source domain and target domain is significantly reduced after domain adaptation operation through our proposed model. This indicates that our model can effectively reduce the difference of data distribution between different sites.

  In the resubmitted manuscript, the relevant presentation has been included in Figure 4 and Section 5.1.

Table S2 Measure the distribution difference between source domain (S) and target domain (T) based on Frobenius norm of the covariance between S and T before and after domain adaptation.

Source (S) → Target (T)

Frobenius norm of the covariance between S and T

Before domain adaptation

NYU→UM

0.0013

NYU→UCLA

0.0094

UM→UCLA

0.0096

After domain adaptation

NYU→UM

0.0010

NYU→UCLA

0.0004

UM→UCLA

0.0002

The added metric analysis in Section 5.1 of the paper is reproduced as follows:

“5.1. Visualization of Data Distribution

......

At the same time, we calculated the Frobenius norm of the covariance (CF) between samples in the source domain and the target domain, which is used to measure the difference of data distribution between different sites. It is observed that the CF between different sites is significantly reduced after domain adaptation. These results show that A2GCN can effectively extract transferable features and reduce domain shift.

The redrawn Figure 4, with its captions, is placed in the new manuscript.

Point 5:  - Figures 6 and 7 can be moved to complementary material or their results can be simply discussed.

Response 5:  Thank you very much for your questions. As suggested, we have now moved the original Figures 6 and 7 in the Supplementary Materials (now Fig. S1 and Fig. S2).

Point 6:  section 5.6 Are authors referring to the potential use of semi-supervised learning? Authors intents need to be clarified.

Response 6: Thank you very much for your questions.

  In this paper, an unsupervised domain adaptation model based on GCN method is established and applied to computer-aided early diagnosis of ASD. In the model training stage, we simultaneously extract the sample features of the source domain with complete data labels and the sample features of the target domain without data labels, and align the features between the domains. Then, in the test phase, the trained classification model is directly applied to the prediction of target domain samples. This is not strictly semi-supervised learning, but transductive learning. The target domain samples without labels are only used as auxiliary feature extraction in the training stage. Unsupervised domain adaptation model is a learning method in which the trained parameters in one model are transferred to a new model for further training and learning. This can be approximated as an unsupervised pre-training method. In the future, we hope to build a more generalized diagnostic model with the help of more unlabeled sample information.

  For clarity, the limitations of this paper and future work have been reiterated in Section 5.6 of the revised paper, which can also be found below.

“5.3. Limitation and Future Work
Although our proposed A2GCN method has achieved good results in the prediction of ASD, there is still challenging work to be considered in the future. First, in our current work, only knowledge transfer between a single source domain and a target domain is considered. It is also interesting to explore the shared features of multiple source domains to reduce the heterogeneity of data and thus improve the learning performance of the target domain. Second, the size of the training sample is relatively small. We hope to add unlabeled samples from other public datasets to assist in pre-training the proposed network in a semi-supervised learning manner, aiming to further improve model generalization capability [49].”

Reviewer 2 Report

The first two paragraphs of introduction do not have comprehensive and coherence. Line 23 has an uncompleted sentence. It is need to write a little bit more in the introduction.

The sequence of tables and figures should be re organized. For example, table 3 was presented so sooner than its explanation in the text. And also, there is need more explanation about figure 1 in the introduction. What did writers mean about target graphs and source graphs? There was some information about them in the next sections, however it should be in the introduction or the figure should be moved to next part.

The result section had too much information and then not easy to understand. It is better to recheck the experiment’s part. The main purpose of this study is to analysis multi center data, however, there is no final result of analyzing ASD with this new method. 

Author Response

Response to Reviewer

We further standardized the English grammar in the paper as suggested. Also, check all references to see if they relate to the content of the manuscript. We have marked the revision track of the paper in LaTeX in blue font so that editors and reviewers can easily view it.

Point 1: The first two paragraphs of introduction do not have comprehensive and coherence. Line 23 has an uncompleted sentence. It is need to write a little bit more in the introduction.

Response: Thank you very much for your advice. We have revised the first two paragraphs of the introduction in the paper. Details can be found in the resubmitted revised version of the paper.

Point 2: The sequence of tables and figures should be re organized. For example, table 3 was presented so sooner than its explanation in the text.

Response: Thank you very much for your reasonable suggestions. We have followed the suggestion to reorganize the pictures and tables and put them in the right place. See the resubmitted revised version of the paper for details.

Point 3: And also, there is need more explanation about figure 1 in the introduction. What did writers mean about target graphs and source graphs? There was some information about them in the next sections, however it should be in the introduction or the figure should be moved to next part.

Response: Thank you for your advice. The source graph refers to the brain functional connectivity networks (FCNs) constructed from the source domain samples, while the target graph refers to the brain FCNs constructed from the target domain samples.

  For clarity, we have given a clearer explanation of Figure 1 in detail in the introduction section. For the convenience of review, we copy the related text below.

“As shown in Figure. 1, we construct a domain adaptation model with attention GCN (A2GCN) of multi-site rs-fMRI for ASD diagnosis. For the convenience of description, we set a known site as the source domain, and define the site to be predicted as the target domain. In this paper, we focus on the classification task of graphs. Therefore, we first construct the corresponding FCNs based on the rs-fMRI data of subjects from the source/target domains, and take the FCNs as the corresponding source/target graphs. Then, we use GCN as a feature extractor to capture the nodes/ROIs representations from the source/target graphs respectively through the graph convolution layers. And the node attention mechanism is applied to explore the contribution weight of nodes/ROIs automatically. Finally, the objective function composed of multiple loss functions is jointly optimized, so as to establish a cross-domain classification model with wider application
range.’’

Point 4: The result section had too much information and then not easy to understand. It is better to recheck the experiment’s part.

Response 4: Thank you very much for your questions. We have reorganized the discussion of the model experiment results in Section 4.4 of the paper, listed as follows.

“4.4. Results

The quantitative results of the A2GCN and several competing methods in ASD vs. HC classification will be reported in Table 3. We observe the following interesting findings.
(1) The four cross-domain classification models (i.e., DNNC, MDD, DANN and A2GCN) achieved better results in most cases compared with several single-domain classification models (i.e., DC, BDC, DNN and GCN). This means that the introduction of domain adaptation learning module helps to enhance the classification performance of the model, which may benefit from the transferable feature representation across sites learned by the model.
(2) Graph-based (i.e., GCN, MDD, DANN and A2GCN)) methods usually produce better classification results than traditional classical methods based on manually defined node features (i.e., DC, BD and BDC) and network embeddings (i.e., DNN and DNNC). Because these traditional methods only consider the characteristics of nodes; however, those methods that use GCN as feature extractor can update and aggregate the features of nodes on the graph end-to-end with the help of the underlying topology information of FCNs, so as to learn more discriminative node representation, which may be more beneficial for ASD auxiliary diagnosis.

(3) The experimental results of the proposed A2GCN consistently outperform all competing methods. This indicates that A2GCN can achieve effective domain adaptation and reduce data distribution differences, thus improving the robustness of the model.
(4) Compared with three advanced cross-domain methods (i.e., DNNC, MDD and DANN), our proposed A2GCN method has a competitive advantage in various domain adaptation tasks. This may be because our method adds node attention mechanism module, which can make intelligent use of different contributions of brain regions. Meanwhile, our method adopts MAE loss and CORAL loss to align different domains step by step. These operations can partially alleviate the negative effects of noisy areas.”

Point 5: The main purpose of this study is to analysis multi center data, however, there is no final result of analyzing ASD with this new method. 

Response 5:Thank you very much for your advice. In this paper, a cross-domain classification model for ASD diagnosis is established for sites with different data distribution (or multi-center sites), and the experimental results of the model are compared with several competing algorithms. The experimental results are shown in Table 3 of the paper.

  In Section 4.4 of the paper, we have further added the analysis of the final prediction results of our model for patients with ASD as suggested. The related text is copied in the following.

“(4) Compared with three advanced cross-domain methods (i.e., DNNC, MDD and DANN), our proposed A2GCN method has a competitive advantage in various domain adaptation tasks. This may be because our method adds node attention mechanism module, which can make intelligent use of different contributions of brain regions. Meanwhile, our method adopts MAE loss and CORAL loss to align different domains step by step. These operations can partially alleviate the negative effects of noisy areas.”

Round 2

Reviewer 1 Report

Authors have answers my concerns

Reviewer 2 Report

The manuscript improved well.